# Impact of Physical Contact on Preterm Infants’ Vital Sign Response to Live Music Therapy

**DOI:** 10.3390/ijerph19159524

**Published:** 2022-08-03

**Authors:** Susann Kobus, Marlis Diezel, Monia Vanessa Dewan, Britta Huening, Anne-Kathrin Dathe, Ursula Felderhoff-Mueser, Nora Bruns

**Affiliations:** 1Department of Paediatrics I, University Hospital, University of Duisburg-Essen, 45147 Essen, Germany; marlis.diezel@gmail.com (M.D.); monia.dewan@uk-essen.de (M.V.D.); britta.huening@uk-essen.de (B.H.); anne-kathrin.dathe@uk-essen.de (A.-K.D.); ursula.felderhoff@uk-essen.de (U.F.-M.); nora.bruns@uk-essen.de (N.B.); 2Centre for Translational Neuro- and Behavioural Sciences (C-TNBS), Faculty of Medicine, University of Duisburg-Essen, 45147 Essen, Germany; 3Department of Health and Nursing, Occupational Therapy, Ernst-Abbe-University of Applied Sciences Jena, 07745 Jena, Germany

**Keywords:** live music therapy, neonatal intensive care unit, physical contact, preterm infants, stabilisation, vital sign response

## Abstract

Evidence that music therapy stabilises vital parameters in preterm infants is growing, but the optimal setting for therapy is still under investigation. Our study aimed to quantify the effect of physical contact during live music therapy in preterm infants born < 32 weeks’ gestational age (GA) on post-therapy vital sign values. Live music therapy was delivered twice-weekly until discharge from hospital to 40 stable infants < 32 weeks’ GA. Baseline and post-therapy heart rate, respiratory rate, oxygen saturation and physical contact during each session were recorded. 159 sessions were performed with, and 444 sessions without, physical contact. Descriptive and multivariable regression analyses based on directed acyclic graphs were performed. The mean GA was 28.6 ± 2.6 weeks, and 26 (65%) infants were male. Mean absolute values for heart and respiratory rates lowered during music therapy regardless of physical contact. The mean post-therapy SaO2 was higher compared to baseline values regardless of physical contact (mean differences −8.6 beats/min; −13.3 breaths/min and +2.0%). There were no clinically relevant changes in vital sign responses between therapy sessions, with or without physical contact, or adjusted post-therapy values for any of the studied vital signs. Physical contact caused better baseline and post-therapy vital sign values but did not enhance the vital sign response to music therapy. Thus, the effect of music therapy on preterm infants’ vital signs is independent of physical contact and parents’ presence during music therapy in the neonatal intensive care unit.

## 1. Introduction

Music therapy is a promising non-medical intervention to reduce stress and pain in preterm and sick newborn infants and children [1,2,3,4]. Preterm infants are repetitively exposed to stressful and painful stimuli during their initial hospital stay. Very preterm birth occurs during a phase of rapid brain development, putting these infants at risk for acquired brain injury from external factors such as stress and physiological instability [5,6]. At the same time, the preterm brain displays great neuroplasticity to compensate possible injury.

Interventions to attenuate stress and stabilize the infant during this crucial phase hold promise to either prevent or attenuate damage to the developing brain. Research investigating long-term effects of music therapy is only evolving [7,8] but vital sign responses to music therapy as a surrogate parameter of short term-stabilisation has been well-studied [9,10,11]. Currently, the optimal setting in which music therapy should be applied is under investigation [12,13,14,15,16].

For example, preterm infants with severe brain injury displayed physiological and behavioural instability during the singing of their mother and kangaroo care [17]. In stable infants with normal hearing, additional live music therapy during kangaroo care to enhance stress reduction showed beneficial effects on heart rate, oxygen saturation [18,19,20,21], length of hospital stay, and re-hospitalisation rate compared to kangaroo care alone [22]. The applied study designs included cross-over designs [19], randomized controlled trials [17,18,21,22], and analysis of a complete cohort [20], typically comparing baseline with post-therapy values.

Physical contact, often delivered as kangaroo care, has long been known to provide various benefits for preterm infants including physiological and behavioural stabilisation and pain relief [23,24,25,26,27]. It attenuates stress in parents, improves parental self-efficacy [28] and reduces pain in routine procedures or examinations in preterm infants [29]. However, the effect of physical contact by touching or holding the infant on infants’ responses to music therapy remains largely unknown.

The aim of this study was to quantify the effect of physical contact on a parent during live-performed music therapy by a qualified music therapist, compared to music therapy alone, on vital sign responses in stable preterm infants born < 32 weeks’ gestational age.

## 2. Methods

### 2.1. Study Design

This study analysed a subset of a prospective randomized controlled clinical trial (Clinical trial number DRKS00025753) that recruited 80 infants for 1:1 randomization to either routine care or routine care plus music therapy. Participating infants were randomly allocated to the intervention group or to the non-intervention group. To investigate the effects of physical contact on vital sign response during music therapy, we analysed protocols of music therapy sessions performed in the intervention group.

### 2.2. Eligibility and Recruitment

Eligible for participation were preterm infants born between October 2018 and May 2021 at the University Hospital Essen < 32 weeks gestational age.

Exclusion criteria were congenital hearing disorders, intraventricular haemorrhage °III according to Papile, periventricular infarction, cerebral malformations, and underlying neurodevelopmental diseases that were known at the time of screening for eligibility for the study. Written informed consent was obtained from the parent/guardian within the first week of life at a minimum age of 72 h for the main study and all associated analyses. The local ethics committee of the Medical Faculty of the University of Duisburg-Essen (18-8035-BO) approved the study. The registration number for clinical research is DRKS00025753.

### 2.3. Intervention

Clinically stable patients received live music therapy from the second week of life twice per week until their hospital discharge [12,30]. The music therapist coordinated the timing of the individual therapy sessions with the nursing staff and parents in compliance with clinical routine. During each session, the infant remained in the same position as it had been before the beginning of the session, either in the incubator, heated cot or parents’ arms. Physical contact was considered as the infant being held by one parent in their arms, on their chest (including kangaroo care), legs or shoulders or if the parent was touching their child with the hand.

During each session the music therapist sang individually improvised melodies and/or used the instrument sansula. In the first few sessions, improvised singing of humming tones or tone sequences was applied and guided by the infant’s breathing and reactions. The sansula has its origins in the African kalimba, which is surrounded by an eardrum. It creates an enveloping and long-lasting soft sound.

Music therapy was planned to last 20 to 30 min and carried out at a low volume for each individual infant in order to not disturb the surrounding patients. Vital signs (heart rate, respiratory rate, and oxygen saturation) were recorded from the patient monitor directly before and at the end of each music therapy session. The presence of parents, and physical contact with the parent during the session were recorded. Clinical data were obtained from the patient history.

### 2.4. Statistical Analyses

Categorical variables are summarized as counts and relative frequencies. Continuous variables were summarised as means with 95% confidence intervals or standard deviation (SD) if normally distributed, and as median with interquartile range if data were skewed. A two-sided t-test was performed to assess vital sign differences between sessions with and without physical contact.

Multivariable analyses were carried out using generalized linear models with the post-therapy values for heart and respiratory rate, and oxygen saturation as outcomes. We used causal diagrams derived from the theory of directed acyclic graphs (DAGs) [31] to define minimally sufficient adjustment sets for each analysis, as recommended for causal inference studies in paediatric and critical care research [32,33]. Based on these DAGs (see Appendix A), we adjusted our analyses by corrected gestational age to estimate the total effect of physical contact on vital signs during music therapy. Critical illness was considered as adjusted per restriction (=study design) because music therapy was performed only in stable infants. Additionally, we adjusted for repeated measurement within individuals. Because it was not required to adjust for baseline vital sign values according to the DAGs, even though this seems natural, we conducted a sensitivity analysis including pre-therapy vital sign values.

All calculations were carried out using SAS Enterprise Guide 8.4 (SAS Institute Inc., Cary, NC, USA).

## 3. Results

### 3.1. Patients

During the study period, 144 infants < 32 weeks’ gestational age received treatment at the University Hospital Essen. Eighty infants were included, 40 in the intervention and 40 in the non-intervention group, which was not further analysed in this study. Sixty-four infants were not recruited (Figure 1). Reasons for non-recruitment (Figure 1) were external referrals (*n* = 3 [5%]), death before recruitment (*n* = 11 [17%]), transfer to an external centre (*n* = 4 [6%]), cerebral haemorrhage °III or higher (*n* = 3 [5%]), critical maternal illness/death (*n* = 2 [3%]), insufficient German language skills to understand the study objectives (*n* = 3 [5%]), hospital stay during the initial phase of the COVID-19 pandemic (music therapy was not permitted during that time) (*n* = 6 [9%]), and refusal (*n* = 32 [50%]). Non-recruited infants had a mean gestational age of 28 + 3 weeks (range 22 + 4 to 31 + 6 weeks) and a mean birthweight of 1151 g (range 210 to 2030 g). Included infants were older and had lower birth weights (Table 1).

### 3.2. Music Therapy Sessions

A total of 604 music therapy sessions were performed between 23 + 6 and 43 + 0 weeks corrected gestational age with a mean duration of 24.2 ± 8.6 min (range 10 to 55 min). One hundred fifty-nine music therapy interventions were conducted with, and 444 interventions without, physical contact with the parents. Each infant received an average of 14 music therapy sessions, of which six were performed with physical contact (range 0 to 19 sessions) and eight without (range 1 to 23 sessions).

### 3.3. Vital Sign Response

Heart and respiratory rates after therapy were lower compared to baseline, while oxygen saturation increased (Table 2, Appendix A). Infants with physical contact during the music therapy session had lower baseline values for heart and respiratory rates and higher baseline oxygen saturations than infants without physical contact. The mean response to music therapy was slightly lower in infants with physical contact compared to infants without physical contact, but nonetheless resulted in better post-therapy values (Table 2, Figure 2). Alike, relative vital sign responses were lower in infants with physical contact compared to no physical contact (Figure 3).

### 3.4. Multivariable Analyses

After adjusting for confounders, there were no clinically relevant differences in vital sign responses between sessions with and without physical contact in the main analyses and the sensitivity analyses (Table 3). The adjusted values yielded no clinically relevant difference from unadjusted values (Table 2 and Table 3). Detailed model information of the main and sensitivity analyses is presented in Appendix A.

## 4. Discussion

This study showed improved vital signs in preterm infants after music therapy regardless of physical contact during the intervention. Baseline values of heart and respiratory rates in our study were lower in infants exposed to physical contact with their parents, resulting in lower absolute post-therapy values. For oxygen saturation, these findings were inverted. After adjustment for confounders, post-therapy vital sign values did not differ between groups.

Only a few studies compared the effect of kangaroo care or physical contact with and without music therapy [20], and music therapy with and without kangaroo care [21], on preterm infants’ vital signs. Combining music therapy with kangaroo care showed stronger declines of heart and respiratory rates and an increase of oxygen saturation compared to kangaroo care alone [21]. Another study with music therapy by a qualified music therapist who sang and played improvised melodies on an ocean disc and guitar showed decreased heart rates and increased oxygen saturations in infants who received music therapy during kangaroo care, while infants receiving music therapy alone showed no vital sign responses [20]. Matching the results of these two studies, combining kangaroo care and lullaby music showed stronger effects on heart rate and oxygen saturation compared to an only-lullaby music group and a control group receiving kangaroo care [34]. In contrast, a recent study found equal effects of music therapy on vital sign responses in infants exposed to physical contact with their parents and those who did not [35]. Our results align with these findings with respect to the relative vital sign responses and the adjusted post-therapy values. Regarding absolute vital sign values, the better baseline values in infants with physical contact before the session resulted in even further stabilisation compared to infants without physical contact. The co-conditioning for regulating heart and respiratory rates, created during gestation, continued after birth during the physical contact, and increased the emotional connection and autonomic co-regulation between mother and infant [36].

According to our findings, physical contact affected baseline vital signs but did not impact the changes induced by music therapy. However, physical contact during music therapy may provide benefits that are not reflected by changes of vital signs, but by physiological stability and possibly neurological functioning. For example, a study on family-centred music therapy found higher weight gain, shorter length of stay, and lower re-hospitalisation rates in preterm infants who received music therapy during kangaroo care compared to no music therapy [22]. Whether combining music therapy with non-medical interventions like non-nutritive sucking, swaddling, positioning, facilitated tucking, kangaroo care or skin-to-skin contact or multi-sensorial stimulation, the practice provides benefits that last beyond short- or mid-term stabilisation and deserves further investigation.

Our study has several limitations. No data on changes in oxygen supplementation during therapy sessions were recorded. Further, music therapy sessions had different durations, were carried out at different times of the day, and with time-lapses from feeding, possibly leading to variations of vital sign responses.

In summary, this study provides evidence that physical contact during music therapy is safe, feasible, and produces similar vital sign responses compared to music therapy without physical contact. Until further investigations have elucidated whether combining physical contact and music therapy provide long-term benefits compared to music therapy alone, music therapy should be integrated into clinical routine, even if parents are absent during the session.

## Figures and Tables

**Figure 1 ijerph-19-09524-f001:**
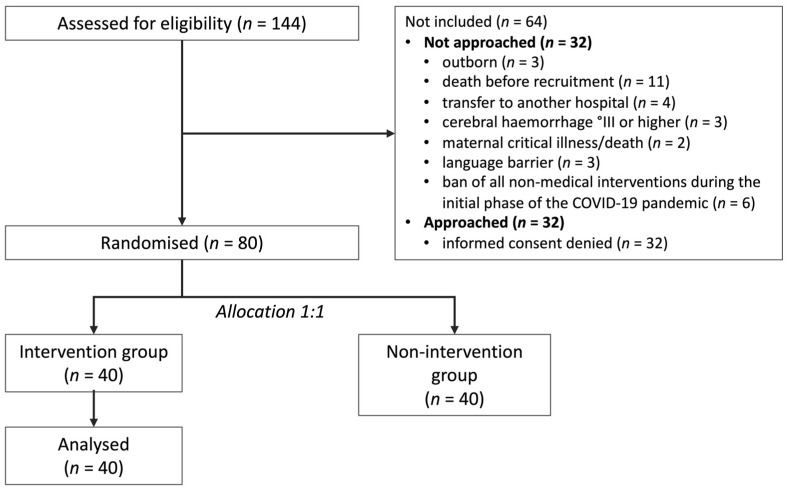
Flow chart of the included and excluded participants of the study.

**Figure 2 ijerph-19-09524-f002:**
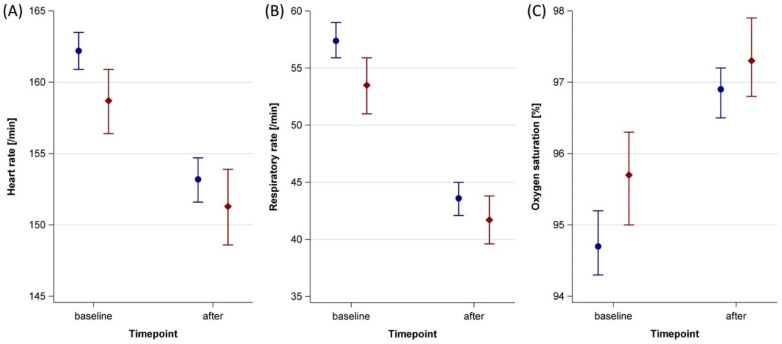
Baseline and post-therapy vital sign values in 40 stable preterm infants receiving music therapy with and without physical contact at the University Hospital Essen between October 2018 and July 2021. Blue = no physical contact (159 sessions); red = with physical contact (444 sessions); error bars = 95 % confidence intervals. (**A**) Heart rate. (**B**) Respiratory rate. (**C**) Oxygen saturation.

**Figure 3 ijerph-19-09524-f003:**
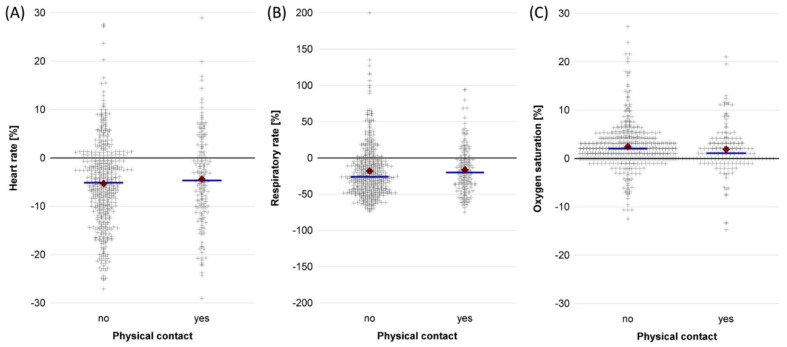
Relative change of vital signs during music therapy sessions in 40 stable preterm infants receiving music therapy with (159 sessions) and without (444 sessions) physical contact at the University Hospital Essen between October 2018 and July 2021. A negative value indicates a lower, and a positive value a higher, post-therapy value compared to the baseline value. + signs = individual measurements; blue line = median; red diamond = mean. (**A**) Heart rate. (**B**) Respiratory rate. (**C**) Oxygen saturation.

**Table 1 ijerph-19-09524-t001:** Clinical characteristics of participants.

	Therapy Group Receiving Music Therapy (*n* = 40)	Non-Intervention Group Not Analysed in This Study (*n* = 40) **
Male [*n* (%)]	26 (65%)	23 (58%)
GA (weeks) [mean ± SD (range)]	28.6 ± 2.6 (23.9–31.7)	28.8 ± 2.5 (22.9–31.9)
Birth weight (g) [mean ± SD (range)]	1136 ± 404 (340–1790)	1147 ± 396 (360–2120)
Died [*n* (%)]	3 (8%)	2 (5%)
Mean age at death (days)	22	78
APGAR score at 10 min [median (IQR)]	9 (8–9)	9 (8–9)
Early-onset sepsis [*n* (%)]	7 (18%)	6 (15%)
Late-onset sepsis [*n* (%)]	10 (25%)	11 (28%)
Bronchopulmonary dypslasia (mild) [*n* (%)]	2 (5%)	2 (5%)
Bronchopulmonary dypslasia (severe) [*n* (%)]	2 (5%)	2 (5%)
Intraventricular haemorrhage °I–II [*n* (%)]	9 (23%)	6 (15%)
Intraventricular haemorrhage °III [*n* (%)] *	2 (5%)	1 (3%)
Necrotizing enterocolitis [*n* (%)]	1 (3%)	2 (5%)
Patent ductus arteriosus [*n* (%)]	17 (43%)	16 (40%)
Medical therapy [*n* (%)]	16 (40%)	16 (40%)
Surgical therapy [*n* (%)]	1 (3%)	0
Antibiotic treatment (days) [median (IQR)]	5 (0–15)	4 (0–6)

APGAR = Appearance, Pulse, grimace, Activity, and Respiration; IQR = interquartile range; SD = standard deviation; * Grade III° haemorrhage developed after inclusion into the study. ** The non-intervention group is presented for transparency to show that randomization produced demographically equal groups.

**Table 2 ijerph-19-09524-t002:** Unadjusted heart rate, respiratory rate and oxygen saturation before and after music therapy sessions with and without physical contact to parents.

	Vital Sign	Sessions (*n*)	Baseline Mean (95% CI)	After Therapy Mean (95% CI)	Difference Mean (95% CI)
Physical contact	Heart rate (beats/min)	159	158.7 (156.4–160.9)	151.3 (148.6–153.9)	−7.4 (−9.8–(−5.1))
	Respiratory rate (breaths/min)	159	53.5 (51.0–55.9)	41.7 (39.6–43.8)	−11.8 (−14.7–(−8.8))
	SaO_2_ (%)	158	95.7 (95.0–96.3)	97.3 (96.8–97.9)	1.7 (1.0–2.3)
No physical contact	Heart rate (beats/min)	444	162.2 (160.9–163.5)	153.2 (151.6–154.7)	−9.0 (−10.3–(−7.7))
	Respiratory rate (breaths/min)	434 *	57.4 (55.9–59.0)	43.6 (42.1–45.0)	−13.9 (−15.8–(−11.9))
	SaO_2_ (%)	446	94.7 (94.3–95.2)	96.9 (96.5–97.2)	2.2 (1.8–2.5)

* Missing values for respiratory rates originate from infants on full mechanical respiratory support during the session.

**Table 3 ijerph-19-09524-t003:** Adjusted heart rate, respiratory rate and oxygen saturation before and after music therapy sessions with and without physical contact to parents.

Vital Sign	Physical Contact	Sessions (*n*)	Adjusted Value after Therapy Mean (95% CI) ^a^	Sensitivity Analysis Mean (95% CI)
Heart rate (beats/min)	yes	159	152.8 (149.2–156.4)	153.4 (150.7–156.1)
	no	444	151.5 (149.2–153.8)	151.8 (149.9–153.8)
Respiratory rate (breaths/min)	yes	159	41.6 (39.4–43.9)	42.2 (39.9–44.4)
	no	434 *	42.6 (40.6–44.6)	42.1 (40.2–44.0)
SaO_2_ (%)	yes	158	97.3 (96.8–97.7)	97.0 (96.5–97.4)
	no	446	97.4 (96.9–97.9)	97.2 (96.9–97.6)

* Missing values for respiratory rates originate from infants on full mechanical respiratory support during the session. ^a^ adjusted for corrected gestational age and repeated measurements within individuals.

## Data Availability

Original data will be made available to any qualified researcher upon request.

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
