# Peer review of "Impact of Physical Contact on Preterm Infants’ Vital Sign Response to Live Music Therapy"

_ijerph, 2022, doi:10.3390/ijerph19159524_

Round 1

Reviewer 1 Report

Kobus et al. present an analysis of infant vital signs in response to music therapy with and without the addition of physical contact. Overall, the study is clearly designed, and the data are mostly clearly presented. The complex comparisons (effect of music with and without physical touch) sometimes obscures the significance of the findings. Some clarifications are suggested:

1. The text refers to supplementary tables (line 123; data shown in Figure 3) and supplementary figures (line 108; data shown in Figures 4-6). Are Figures 3-6 supposed to be supplemental figures, or are they part of the main text? If so, please re-label as needed.

2. Figures 4-6 (directed acyclic graphs) would be more helpful for the reader with more details, either in the legends or section 2.4. What is the significance of the color coding? Why are "individual characteristics" circled in some graphs but a rectangle in others? How were the significant parameters identified?

3. If this reviewer understands the patient selection correctly, 80 of 144 infants were included in the study. Forty of these patients underwent music therapy with and without physical contact. What is the purpose of the 40 "control" patients reported in Table 1? It appears that all subsequent data only represent the 40 patients in the music therapy group, who served as their own controls when they had music therapy without their parents present.

4. In Table 2, please consider only presenting the separate data for "Physical contact" and "No physical contact". The significance of including the overall data are unclear, given the hypothesis the physical contact will make a difference. Additionally, are the baseline values statistically different between the "physical contact" and "no physical contact" datasets?

5. There appears to be a typo on line 189 ("compared to a- only-lullaby music group").

Author Response

Dear Reviewer,

thank you for giving us the opportunity to improve our manuscript entitled "Impact of physical contact on preterm infants’ vital sign response to live music therapy".

Please find below our revisions addressed in point-by-point fashion. Changes to the original manuscript were marked up using the “Track Changes” function.

Kobus et al. present an analysis of infant vital signs in response to music therapy with and without the addition of physical contact. Overall, the study is clearly designed, and the data are mostly clearly presented. The complex comparisons (effect of music with and without physical touch) sometimes obscures the significance of the findings. Some clarifications are suggested:

COMMENT 1

The text refers to supplementary tables (line 123; data shown in Figure 3) and supplementary figures (line 108; data shown in Figures 4-6). Are Figures 3-6 supposed to be supplemental figures, or are they part of the main text? If so, please re-label as needed.

Thank you for pointing this out. We rearranged the order of the figures in the main text and relabelled figures 4-6 as supplementary figures S1-S3.

COMMENT 2

Figures 4-6 (directed acyclic graphs) would be more helpful for the reader with more details, either in the legends or section 2.4. What is the significance of the color coding? Why are "individual characteristics" circled in some graphs but a rectangle in others? How were the significant parameters identified?

We added explanations of the colour coding to the legends and corrected the rectangle of individual characteristics in figure S1.

The theory of DAGs is quite complex. Briefly, a DAG consists of edges and vertices that strictly follow a forward flow from the exposure towards the outcome (no circle relationships = acyclic). Multiple paths can exist, and the minimally sufficient adjustment set is a set of variables that “blocks” all paths. This set can be identified manually or by software. We used the freely available DAGitty software (http://www.dagitty.net).

COMMENT 3

If this reviewer understands the patient selection correctly, 80 of 144 infants were included in the study. Forty of these patients underwent music therapy with and without physical contact. What is the purpose of the 40 "control" patients reported in Table 1? It appears that all subsequent data only represent the 40 patients in the music therapy group, who served as their own controls when they had music therapy without their parents present.

That’s correct. The control group is presented to show that randomization produced groups with equal demographic characteristics. However, we deleted the “overall” column as it did not provide information relevant for the present study.

Also, we added this information to the table footnote: The control group is presented for transparency to show that randomization produced demographically equal groups.

COMMENT 4

In Table 2, please consider only presenting the separate data for "Physical contact" and "No physical contact". The significance of including the overall data are unclear, given the hypothesis the physical contact will make a difference. Additionally, are the baseline values statistically different between the "physical contact" and "no physical contact" datasets?

We removed the overall data from the table.
This study aimed to estimate effect rather than assess the data for statistical significance. The reason is, that with increasing case number, statistical significance can be reached even for very small effects that may not be clinically relevant. Therefore, the focus of this paper was to quantify effects. However, an indirect measure to deduce statistical significance are non-overlapping confidence intervals (CI). The presented baseline CIs in table 2 are almost non-overlapping for heart and respiratory rate but overlapping for SaO2.

COMMENT 5

There appears to be a typo on line 189 ("compared to a- only-lullaby music group").

We corrected the typing error in line 189.

Reviewer 2 Report

Very interesting study and most definitely a needed piece of research. The findings are interesting and highlight how music in itself can be beneficial for pre-term infants. 

In the Introduction section, it would be interesting to read about about other previous studies in more detail. In particular, a mention on previous study designs and methods would be helpful. 

In the Discussion section, it would be interesting to read about comparative results at greater length; however, we understand that there might not be space in this article to do this. 

The graphs and results are presented in a very comprehensive and easy-to-undertand manner. 

Author Response

Dear Reviewer,

thank you for giving us the opportunity to improve our manuscript entitled "Impact of physical contact on preterm infants’ vital sign response to live music therapy".

Please find below our revisions addressed in point-by-point fashion. Changes to the original manuscript were marked up using the “Track Changes” function.

Very interesting study and most definitely a needed piece of research. The findings are interesting and highlight how music in itself can be beneficial for pre-term infants. 

The graphs and results are presented in a very comprehensive and easy-to-undertand manner. 

COMMENT 1

In the Introduction section, it would be interesting to read about other previous studies in more detail. In particular, a mention on previous study designs and methods would be helpful. 

We agree that this would be very interesting, especially because research in music therapy is evolving rapidly and there are many different methods being applied and new methods being developed. However, a review of current study designs, their differences, and subsequent implications for interpretation would require an article on its own. However, we added the following sentence to point out that there a variety of approaches is used:

The applied study designs included cross-over designs [19], randomized controlled trials [18, 21, 22, 23], and analysis of a complete cohort [20], typically comparing baseline with post therapy values.

COMMENT 2

In the Discussion section, it would be interesting to read about comparative results at greater length; however, we understand that there might not be space in this article to do this. 

We agree that this would be very interesting but we are afraid that this would be a topic suitable for a comprehensive literature review.

Reviewer 3 Report

Thank you for your paper on the impact of physical contact on preterm babies.  It is worth reading the article. I have some comments on the paper.

1. The abstract is structured and concise. In the conclusion section of your abstract, you have concluded that music therapy can be delivered independently... However, from the title of your study, it looks like your study is to evaluate the effect of physical contact rather than music therapy. Therefore, I suggest amending the final statement on the abstract.

2. Background:

2.1 Is there any systematic evidence on the impact of music therapy on the vitals of preterm or term infants (any systematic reviews or meta-analysis)?

2.2 The authors have discussed on music therapy and Kangaroo care in the background. However, from the final paragraph of the background, it is clear that the authors aim to quantify the effect of physical contact. It would be important to discuss background evidence on physical contact and vital signs... either in preterm or term infants and postulate what is the existing gap in the literature. The current background is comparing apples with oranges which needs to be amended.

3.  The study is a subset of a study to evaluate the effect of routine care vs routine care plus music therapy. How can the authors fulfil the objective to evaluate the impact of physical touch when the study was not conducted with this objective. The sample size estimation with one theory cannot be applicable to others and hence the statistical analysis cannot be projected to this research.

4. In section 2.4 the authors mention that they adjusted the analysis to estimate the total effect of physical contact on vital signs on music therapy where as in the results they have included neonates admitted under the control arm. The authors should clarify how they analysed the participants in the control group.

5. The authors should justify why they included the participants from the control group (who did not receive music therapy) for analysis.

6. The authors should give the characteristics of excluded participants. Were they similar to the included participants or were they different? The authors have explained two basic demographic characteristics (lines 128-130) but there are more than 10 characteristics in table 1. As almost 50% of the participants are excluded, it can impact the findings if the characteristics of the excluded participants are different from those included in the study.

7. If the authors have not analysed the control group in this paper, it is advisable to remove the information in the text and in charts as it is confusing to the readers. Please review the above points. It is, therefore, important to avoid confusion. Table 1 is Ok. The authors can put a footnote in table 1 mentioning that they have not included these 40 participants in this manuscript rather than making it confusing to the readers.

8. In Line 138, the authors have mentioned the number of music therapies with and without physical contact. So, I would suggest to adhere this information in the paper clearly and remove control group participants in the manuscript to make it clear.

9. Likewise, as mentioned earlier, it is not advisable to comment on the statistical significance of any parameters from this study as the sample size was not determined from this perspective (neither the number of participants or the number of sessions with or without contact). It should be elaborated on methodological limitations and suggestions be made to conduct this study with a proper sample size estimation to determine if these interventions are statistically significant. 

10. First line of discussion (line 174)- how can we say that vital signs improved with music therapy when we have not analyzed participants in the control group?

11. Line 182 - the word 'decreases' should be replaced with a suitable one.

12. Line 199. We cannot conclude but can only suggest as the sample size estimation to determine this effect was not calculated. 

13. One important ethical question - was the consent obtained for the original clinical trial or for this present analysis (study). It has to be clarified in the ethics section. Did the participants agree to subgroup analysis or to use their information for a separate study? It has to be clarified.

Author Response

Dear Reviewer,

thank you for giving us the opportunity to improve our manuscript entitled "Impact of physical contact on preterm infants’ vital sign response to live music therapy".

Please find below our revisions addressed in point-by-point fashion. Changes to the original manuscript were marked up using the “Track Changes” function.

Thank you for your paper on the impact of physical contact on preterm babies.  It is worth reading the article. I have some comments on the paper.

COMMENT 1

The abstract is structured and concise. In the conclusion section of your abstract, you have concluded that music therapy can be delivered independently... However, from the title of your study, it looks like your study is to evaluate the effect of physical contact rather than music therapy. Therefore, I suggest amending the final statement on the abstract.

We changed into: Thus, the effect of music therapy on preterm infants’ vital signs is independent of physical contact and parents’ presence during music therapy in the neonatal intensive care unit.

COMMENT 2

Background:

Is there any systematic evidence on the impact of music therapy on the vitals of preterm or term infants (any systematic reviews or meta-analysis)?

Several studies showed that music therapy has a stabilizing on vital signs, including a systematic review and a meta-analysis:

  • Bieleninik Ł., Ghetti C., Gold C. Music Therapy for Preterm Infants and Their Parents: A Meta-analysis. 2016;138:e20160971. doi: 10.1542/peds.2016-0971.
  • Costa, V. S., Bündchen, D. C., Sousa, H., Bündchen Pires, L., & Felipetti, F. A. (2022). Clinical benefits of music-based interventions on preterm infants' health: A systematic review of randomised trials. Acta Paediatr. 2022 Mar;111(3):478-489. doi: 10.1111/apa.16222.

These papers are both cited in our manuscript.

The authors have discussed on music therapy and Kangaroo care in the background. However, from the final paragraph of the background, it is clear that the authors aim to quantify the effect of physical contact. It would be important to discuss background evidence on physical contact and vital signs... either in preterm or term infants and postulate what is the existing gap in the literature. The current background is comparing apples with oranges which needs to be amended.

The type of physical touch applied in our study included kangaroo care, holding the infant using other “techniques”, and mere touch with the hand. Kangaroo care and physical touch itself have been proven to stabilize preterm infants, e.g., via increased heart rate variability. We revised the section about kangaroo for clarity added according citations on the effects of kangaroo care and physical touch.

We also clarified in the methods section that kangaroo care was included.

COMMENT 3

The study is a subset of a study to evaluate the effect of routine care vs routine care plus music therapy. How can the authors fulfil the objective to evaluate the impact of physical touch when the study was not conducted with this objective. The sample size estimation with one theory cannot be applicable to others and hence the statistical analysis cannot be projected to this research.

Sample size estimation is performed to calculate the required number of participants to test a hypothesis based on certain assumptions and pre-existing knowledge. The aim of this study, however, was to estimate the strength of effect rather than to test a hypothesis.

For that reason, we applied methods that quantify the effect of physical contact on vital sign responses. We did not perform analyses on statistical significance, even though statistical significance can indirectly be deduced if confidence intervals do not overlap.

COMMENT 4

In section 2.4 the authors mention that they adjusted the analysis to estimate the total effect of physical contact on vital signs on music therapy where as in the results they have included neonates admitted under the control arm. The authors should clarify how they analysed the participants in the control group.

The control group is presented to show that randomization produced groups with equal demographic characteristics. However, we deleted the “overall” column as it did not provide information relevant for the present study.

Also, we added this information to the table footnote: The control group is presented for transparency to show that randomization produced demographically equal groups.

COMMENT 5

The authors should justify why they included the participants from the control group (who did not receive music therapy) for analysis.

See comment above. The control group was not analyzed further.

COMMENT 6

The authors should give the characteristics of excluded participants. Were they similar to the included participants or were they different? The authors have explained two basic demographic characteristics (lines 128-130) but there are more than 10 characteristics in table 1. As almost 50% of the participants are excluded, it can impact the findings if the characteristics of the excluded participants are different from those included in the study.

Yes, the excluded patients are different from the included patients by study design, because patients with certain conditions were not eligible. Further, immortal time bias was introduced into the group of recruited patients by the fact that recruitment started only at 72 hours of life, and a patient who died before 72 hours could not have been included.

We would be very happy to present more detailed information on non-recruited patients. However, these infants’ parents/guardians did not sign written informed consent. According to local legislation it is prohibited to collect more than this minimum amount of information prospectively without informed consent.

Details on the reasons for exclusion are presented in the flow chart (figure 1) to give the reader an overview about potential imbalances between the cohorts.

COMMENT 7

If the authors have not analysed the control group in this paper, it is advisable to remove the information in the text and in charts as it is confusing to the readers. Please review the above points. It is, therefore, important to avoid confusion. Table 1 is Ok. The authors can put a footnote in table 1 mentioning that they have not included these 40 participants in this manuscript rather than making it confusing to the readers.

We revised table 1 (see answers to comments 4 and 5).

COMMENT 8

In Line 138, the authors have mentioned the number of music therapies with and without physical contact. So, I would suggest to adhere this information in the paper clearly and remove control group participants in the manuscript to make it clear.

We revised the second sentence of the results section: Eighty infants were included, 40 in the intervention and 40 in the control group, which was not further analysed in this study.

This information is important because otherwise it will be difficult to explain the group of non-recruited patients.

COMMENT 9

Likewise, as mentioned earlier, it is not advisable to comment on the statistical significance of any parameters from this study as the sample size was not determined from this perspective (neither the number of participants or the number of sessions with or without contact). It should be elaborated on methodological limitations and suggestions be made to conduct this study with a proper sample size estimation to determine if these interventions are statistically significant. 

That is correct, we revised the two sentences that commented on statical significance. Now they only state that were no clinically relevant differences.

COMMENT 10

First line of discussion (line 174)- how can we say that vital signs improved with music therapy when we have not analyzed participants in the control group?

We revised the sentence to be less suggestive of causation: This study shows improved vital signs in preterm infants after music therapy regardless of physical contact during the intervention.  

COMMENT 11

Line 182 - the word 'decreases' should be replaced with a suitable one.

We replaced the word by “declines”.

COMMENT 12

Line 199. We cannot conclude but can only suggest as the sample size estimation to determine this effect was not calculated. 

We found similar vital sign responses between sessions with and without physical contact by applying methods to estimate effects. Further, the sentence is attenuated by the introductory phrase “According to our findings …”. We prefer to leave it as it is.

COMMENT 13

One important ethical question - was the consent obtained for the original clinical trial or for this present analysis (study). It has to be clarified in the ethics section. Did the participants agree to subgroup analysis or to use their information for a separate study? It has to be clarified.

We revised the ethics statement accordingly: Written informed consent was obtained from the parents or legal guardians for the main study and all associated analyses before inclusion into the study.

Reviewer 4 Report

Dear authors, congratulations for the text; it seems to me a very interesting subject raised with an appropriate methodological approach. I suggest for a next study to investigate the influence of attachment styles.

Author Response

Dear Reviewer,

thank you very much for your review to our manuscript entitled "Impact of physical contact on preterm infants’ vital sign response to live music therapy".

Reviewer 5 Report

Authors have done an outstanding job of submitting a publication-ready manuscript. I would recommend accepting it for publication. 

Author Response

(The authors gave the same response as above.)
